# Comparison between the Astaxanthin Release Profile of Mesoporous Bioactive Glass Nanoparticles (MBGNs) and Poly(3-hydroxybutyrate-*co*-3-hydroxyvalerate) (PHBV)/MBGN Composite Microspheres

**DOI:** 10.3390/polym15112432

**Published:** 2023-05-24

**Authors:** Arturo E. Aguilar-Rabiela, Shahin Homaeigohar, Eduin I. González-Castillo, Mirna L. Sánchez, Aldo R. Boccaccini

**Affiliations:** 1Institute of Biomaterials, Department of Materials Science and Engineering, University of Erlangen-Nuremberg, Cauerstrasse 6, 91058 Erlangen, Germany; mirna.sanchez@fau.de; 2Tissue Engineering Research Group, Department of Anatomy & Regenerative Medicine, Royal College of Surgeons in Ireland (RCSI), D02 YN77 Dublin, Ireland; 3School of Science & Engineering, University of Dundee, Dundee DD1 4HN, UK; shomaeigohar001@dundee.ac.uk; 4Polymer Institute, Slovak Academy of Sciences, Dúbravská Cesta 9, 845 41 Bratislava, Slovakia; eduin.ivan@gmail.com; 5AO Research Institute Davos, Clavadelerstrasse 8, 7270 Davos, Switzerland; 6Laboratorio de Farmacología Molecular, Departamento de Ciencia y Tecnología, Universidad Nacional Quilmes, Bernal B1876, Argentina

**Keywords:** Astaxanthin, composite microspheres, sustained release, controlled release

## Abstract

In recent years, composite biomaterials have attracted attention for drug delivery applications due to the possibility of combining desired properties of their components. However, some functional characteristics, such as their drug release efficiency and likely side effects, are still unexplored. In this regard, controlled tuning of the drug release kinetic via the precise design of a composite particle system is still of high importance for many biomedical applications. This objective can be properly fulfilled through the combination of different biomaterials with unequal release rates, such as mesoporous bioactive glass nanoparticles (MBGN) and poly(3-hydroxybutyrate-*co*-3-hydroxyvalerate) (PHBV) microspheres. In this work, MBGNs and PHBV-MBGN microspheres, both loaded with Astaxanthin (ASX), were synthesised and compared in terms of ASX release kinetic, ASX entrapment efficiency, and cell viability. Moreover, the correlation of the release kinetic to phytotherapeutic efficiency and side effects was established. Interestingly, there were significant differences between the ASX release kinetic of the developed systems, and cell viability differed accordingly after 72 h. Both particle carriers effectively delivered ASX, though the composite microspheres exhibited a more prolonged release profile with sustained cytocompatibility. The release behaviour could be fine-tuned by adjusting the MBGN content in the composite particles. Comparatively, the composite particles induced a different release effect, implying their potential for sustained drug delivery applications.

## 1. Introduction

Natural compounds have gained a high interest in recent years due to their beneficial properties, which are being continuously discovered year by year [1,2,3]. In this context, ASX is a well-studied phytotherapeutic reported for its remarkable antioxidant activity [4,5,6]. ASX is known as an xanthophyll carotenoid and was for the first time extracted from lobster by Kuhn and Sorensen [7] and later introduced into the market as a pigmentation material for the aquatic farm industry [8,9]. ASX, 3,3′-dihydroxy-β,β-carotene-4,4′-dione, is classified as a tetraterpene comprising 40 carbon atoms, with the molecular formula of C_40_H_52_O_4_ and a molecular mass of 596.85 g/mol [10,11].

In recent years, ASX has drawn extensive attention for dermatological and cosmetic applications due to its outstanding antioxidant activity, which prevails over that of tocopherol, its supportive effect on skin health, and its protective role against UV radiation [12,13]. However, ASX is poorly soluble in water and is chemically unstable, and shows insufficient bioavailability, challenging its cosmetic and pharmaceutical applications [14]. The creation of delivery systems that can protect ASX and optimise its bioavailability is thus a highly demanded research topic. Regarding drug delivery systems, nanoparticles can be considered a proper candidate to carry drugs and release them locally for various medical applications [15,16]. However, safe assimilation and minimising the side effects of the administrated nanoparticle carriers are still existing challenges [17,18]. In this regard, mesoporous bioactive glass nanoparticles (MBGNs) have been proven to be bioactive and biocompatible drug carriers and, thus, have been proposed for various biomedical purposes [19,20,21]. Moreover, they can release certain ions upon dissolution that can drive complex gene transduction pathways in cells, thereby improving cell activities [22,23]. Despite such merits, there is a need to further extend the time span of drug release from MBGNs in a controlled manner [24]. In this regard, the development of particle carrier systems based on a formulation composed of biodegradable biopolymers, e.g., polyhydroxyalkanoates (PHAs), and MBGNs has been explored as a proper strategy [24,25]. PHAs are a family of biotechnology-produced polymers frequently used for biomedicine, thanks to their outstanding biocompatibility and biodegradability [23]. Particularly, poly(3-hydroxybutyrate) (P(3HB)) and poly(3-hydroxybutyrate-*co*-3-hydroxyvalerate) (PHBV) are well-studied types of PHAs that have been largely employed for the development of drug delivery microspheres [26,27,28,29]. Compared to P(3HB), PHBV has a lower melting temperature and is partially amorphous, thus is considered a more suitable candidate for drug delivery purposes [30,31]. Li et al. [24], for instance, developed vancomycin-loaded PHBV microspheres for coating 45S5 BG bone-tissue-engineering scaffolds. Different PHBV-based microsystems used for drug delivery microsystems have been reported for the release of some phytotherapeutics and natural components for different objectives [32,33,34]. In the current study, ASX-loaded MBGNs are incorporated in PHBV microspheres to extend the ASX release profile of the drug delivery system compared to when ASX is solely loaded into MBGNs. This composite strategy is assumed to enhance cell activities due to the controlled release of both ASX and beneficial ions from MBGN and dissolution byproducts from the MBGN/PHBV composite material.

## 2. Materials and Methods

### 2.1. Materials

PHBV was purchased from Goodfellow (Huntington, UK), dichloromethane and Astaxanthin (ASX) from Sigma-Aldrich (Steinheim, Germany), and Polyvinyl Alcohol (PVA) from Baxter Healthcare (Opfikon, Switzerland). Tetraethyl orthosilicate (TEOS; 99%), triethyl phosphate (TEP; 99%), and calcium nitrate were bought from Aldrich (Steinheim, Germany). Furthermore, ethyl acetate and cetyl-trimethylammonium bromide (CTAB) were provided by Merck (Darmstadt, Germany), ammonium hydroxide (28%) was provided by VWR (Fontenay Sous Bois, France), and distilled water (MilliQ) and absolute ethanol (99.8%) were provided by Alfa Aesar (Kandel, Germany). All the used chemicals were of analytical grade.

### 2.2. Synthesis of MBGN and Composite Microspheres

The mesoporous bioactive glass nanoparticles (MBGN) were produced following a modified Stöber procedure, which was previously reported in [35]. For the loading of ASX into MBGNs, a technique based on previous reports was designed [21,32]. Briefly, 2 mg of synthesised MBGN and 20 μg of ASX were suspended on 5 mL of non-ASX solvent (ultrapure water) and stirred at 800 rpm. Then, to facilitate the incorporation of ASX into the MBGN, the suspension was centrifuged at 8000 rpm. The supernatant was carefully removed to let the precipitated MBGN be loaded with ASX at 10% (*w*/*w*) and dried for 6 h in an incubator at 60 °C.

Simultaneously, composite PHBV-MBGN-ASX microspheres were produced using a solid-in-oil-on-water (S/O/W) method, which was previously reported in [21,32]. Briefly, 2 mg of the synthesised MBGN and 20 μg of ASX were added to a 10 mL of PHVB-DCM solution and stirred at 800 rpm to produce S/O phase. In parallel, an aqueous solution of 1 mg/mL of PVA was prepared and mixed at 600 rpm to form the W phase. Afterward, the S/O phase was dropwise added to the W phase while homogenising at 19,000 rpm using a T18 (IKA, Staufen, Germany) to produce composite microspheres comprised of 10% of MBGN and 1% of ASX (*w*/*w*) of loading. There was a PHBV/MBGN ratio of (10:1) for all the samples. Subsequently, the emulsion was centrifuged at 6000 rpm, and the supernatant was removed entirely to let the precipitated microspheres dry overnight in an incubator at 60 °C. All the particle synthesis was carried out in the beakers and tubes protected against light.

### 2.3. Particle Size and Zeta Potential

The average particle diameter and zeta potential measurement of the Astaxanthin-loaded MBGN (MBGN/ASX), the PHBV-MGBN-ASX (composite microspheres), blank MBGN, and blank PHBV particles were carried out by using a Zetasizer Nano ZS (Malvern, Worcestershire, UK) in deionized water for MBGN conditions and ethanol for PHBV conditions, similar to previous reports [21,36].

### 2.4. Surface Morphology of Particles

Micrographs of the surface morphology of the MBGN/ASX and the composite microspheres were obtained via Scanning Electron Microscopy (SEM) using an SEM Auriga (Zeiss, Munich, Germany). All samples were sputtered with gold in a turbomolecular-pumped coater Q150T Plus (Quorum, Laughton, UK) prior to SEM examination.

### 2.5. Chemical-Structural Characterization of Particles

Absorbance infrared spectra were obtained for the neat components, MBGN/ASX, and composite microspheres by using a Fourier Transform Infrared spectrometer (FTIR) Nicolet 6700 (Thermo-Scientific, Waltham, MA, USA). Measurements were carried out in the middle-infrared (MIR) region from 4000 to 400 cm^−1^ (40 scans and 4 cm^−1^ spectral resolution).

### 2.6. Astaxanthin Entrapment Efficiency

The ASX Entrapment Efficiency (EE) was calculated according to the supernatant method previously reported [37]. The ASX-MBGN and composite microspheres were immersed in 10 mL of ethanol, respectively, and after release depletion, the supernatant concentration was measured using a UV/Vis spectrophotometer Specord 250 (Analytik-jena, Jena, Germany) at λ = 478 nm [38]. The entrapment efficiencies were calculated by using the initially added ASX amounts during the fabrication of particles, the concentration value was measured via UV/Vis, and the following equation was obtained [21,37]:(1)$$EE=\frac{{(M}_{the}-M_{sup})\cdot 100\%}{M_{the}}$$
where M_sup_ is the ASX mass measured in the supernatant in mg, and M_the_ is the initial ASX mass added during the fabrication in μg.

### 2.7. Astaxanthin Release Kinetics

The ASX-release kinetics were calculated by adding 5 mg of each sample to 5 mL of ethanol in triplicate [21,37]. The release kinetic curve was obtained through a technique previously reported [21,37,39]. Briefly, several measurements of the supernatant concentration were taken along the release time by using a Specord 250 device (Analytik Jena, Jena, Germany) at 478 nm. Concentration curves were normalized with respect to the calculated theoretical drug loaded during the fabrication of the samples, and all release kinetics were carried out in the tubes protected from light prior to each measurement. Calibration curves for ASX were obtained prior to each release kinetic assay.

### 2.8. Cell Viability Assays

NIH-3T3 mouse fibroblasts cells (Sigma-Aldrich, Germany) were grown in Dulbecco’s Modified Eagle Medium (DMEM) supplemented with BCS, sodium pyruvate, and L-glutamine (Gibco, Dreieich, Germany). Afterward, 50 × 10^3^ cells were taken from 90% confluency flasks and seeded in 24-well plates. Considering the EE of each condition, master solutions of ASX/MBGN, composite microspheres, and free ASX were so prepared to have a similar content through each condition. Afterward, 10-fold serial dilutions of 10 µL, 100 µL, and 1000 µL from each master solution were added to the 24-well plates previously seeded with cells. A similar amount of blank MBGN and blank PHBV microspheres was used as additional controls, and untreated cell samples were used as control. All experiments were carried out in triplicate. The cell viability was determined via WST-8 assay (Sigma-Aldrich, Steinheim, Germany) after 24 and 72 h post-administration according to the provider protocol. Briefly, the supernatant was carefully removed from all well plates and washed with HBSS; later, freshly prepared cell culture medium containing 1% *v*/*v* WST-8 solution was added to the 24-well plates and incubated for 3 h. After incubation, 100 µL of supernatant from each well plate was transferred, by duplicates, into a 96-well plate for the absorbance measurement at λ = 450 nm in a PHOmo Autobio (Labtec Instruments, Zhengzhou, China), similar to previous reports [21].

### 2.9. Cell Staining and Imaging

Cell samples were stained with Life & Dead Hoechst protocol using Calcein AM, Propidium Iodide, and Life & Dead assays (Merck, Darmstadt, Germany) for cell morphology observation after 72 h of incubation. Cell staining was carried out according to the provider’s protocol. After staining, the samples were stored and protected from light prior to observations. Fluorescent micrographs were obtained and processed using a fluorescent optical microscope Primovert with AxioCam ERc 5s (Zeiss, Munich, Germany).

### 2.10. Bioactivity Assessment

MBGN/ASX and composite microsphere samples were immersed in SBF, produced according to the protocol of Kokubo [40], and incubated for 14 days. Afterward, SBF was removed, and the samples were dried in an incubator at 60 °C for 4 h, then stored at room temperature prior to examination.

The surface morphology of the samples was analysed via SEM according to Section 2.4. The X-ray diffraction analyses (XRD) were performed by using a Rigaku MiniflexTM 600 (Rigaku, Tokyo, Japan) X-ray diffractometer, provided with a Cu-Kα radiation (λ = 1.5406 Å) X-ray source and scintillation counter (Kβ filter) detector, in the angular range (2θ) from 10° to 40° with a resolution of 0.01° and speed scanning of 1°/min.

### 2.11. Statistics

Experiments were carried out in triplicate. The analysis of statistical significance was conducted by analysis of variance One-Way ANOVA followed by the Tukey test (*p* < 0.05), using the software Origin 8 (OriginLab, Northampton, MA, USA), and the data were presented as mean ± standard deviation for each measurement.

## 3. Results and Discussion

### 3.1. Particle Size, Morphology, and Zeta Potential

The synthesised MBGN/ASX nanoparticles and PHBV/MBGN/ASX microspheres exhibited a homogeneous, spherical shape (Figure 1d,e). In contrast to the composite microspheres, which possessed quasi-smooth surface morphology with low porosity (Figure 1e), MBGN/ASX particles featured a mesoporous surface, as similarly reported by Nawaz et al. [35]. The spherical shape of the composite microspheres was not affected by either the incorporation of MBGN or ASX. In both formulations, the spherical shape and surface morphology can promote a homogeneous drug release by diffusion, as observed in similar systems [37,41,42]. The particle size distributions of MBGN/ASX and the composite microspheres are shown in Figure 1a and Figure 1b, respectively. Both systems were monodisperse in terms of particle size distribution, with a diameter between 400 to 600 nm for MBGN/ASX and 2 to 6 μm for the composite microspheres. The size difference, Figure 1f, might affect the ASX release rate due to the non-comparable surface area to volume ratio of the particle carriers [21,41]. The zeta potential of MBGN/ASX nanoparticles and composite microspheres is shown in Figure 1c. The values showed moderate stability, which may reduce the probability of undesired agglomeration for in vivo and in vitro applications [43,44]. Similar Zeta potential values have been observed in previous formulations using MBGN- and PHBV-based particles [21,45,46].

### 3.2. Composition Analysis

The FTIR absorbance spectra of the neat components (ASX, MBGN, and PHBV), MBGN/ASX, and composite microspheres recorded in the mid-infrared region (MIR) from 4000 to 400 cm^−1^ are shown in Figure 2. Three characteristic bands associated with ASX are observed at 1654, 1552, and 974 cm^−1^ in the MBGN/ASX spectrum, which are assigned, respectively, to the C=O stretching vibration, the stretching vibration of C=C in the hexatomic ring, and C–H in the C,C conjugate [47,48]. Except for the band at 1654 cm^−1^, which is overlapped with that of PHBV, the other two characteristic bands of ASX are readily distinguished in the FTIR spectrum of the composite microspheres. Additionally, in the FTIR spectrum of the composite microspheres in Figure 2, the band at 1100 cm^−1^ corresponds to the Si–O–Si bending mode of the MBGN [21]. The bands at 1282 and 1179 cm^−1^ correspond to the C–O–C stretching modes of the crystalline and amorphous parts of PHBV, while the band peak at 1731 cm^−1^ is ascribed to the C=O crystalline phase of PHBV [21], which is overlapped with the carbonyl vibration in ASX. as has been previously indicated. The FTIR spectra confirm the successful loading of ASX in MBGN/ASX and the incorporation of ASX and MBGN in the composite microspheres, which may overlap with the 1654 cm^−1^ band peak of MBGN. These results support the loading of ASX in MBGN/ASX and the integration of ASX and MBGN in the composite microspheres.

### 3.3. Entrapment Efficiency and Release Kinetics

The ASX entrapment efficiencies (EE) of the MBGN/ASX and composite microspheres are shown in Figure 3a. The EE was calculated to be 60% for the MBGN/ASX and 80% for the composite microspheres. This difference could arise from the additional ASX entrapped into the PHBV polymer matrix, improving the EE of the composite in comparison with the open mesoporous structure of the MBGN/ASX, which may facilitate the diffusion of the phytotherapeutic agent through the porous structure and reduce the loading capacity. The ASX release kinetics of MBGN/ASX and the composite microspheres in an organic solvent (ethanol) are compared in Figure 3b, in which the final concentration of ASX released is comparable for both particle systems (~4 μg/mL).

The cumulative release with respect to the initial theoretical amount of ASX in both drug delivery systems exhibited a characteristic biphasic controlled release curve. Nonetheless, the final concentration in MBGN/ASX was slightly lower than that in the composite microspheres, a behaviour that corresponds to the EE difference between the systems. Additionally, a steeper slope was observed in the ASX-MBGN kinetic, which is related to a higher release ratio during the burst phase and resulted in an apparent complete depletion of the release during the first 40 h. In contrast, the composite release kinetic exhibited a lower slope during the burst phase and an extended controlled phase up to the first 80 h after administration, allowing a prolonged ASX release. This dissimilarity in the release behaviour may be explained based on the morphology and composition features of the systems, as the MBGN exhibited a mesoporous surface that favours the release as soon as the nanoparticles are in contact with the medium. In addition, the surface contact area/volume ratio is higher in the MBGNs because of their nanometric dimensions, promoting a fast drug release.

On the contrary, in the composite microspheres, the surface contact area/volume ratio is low. Furthermore, the homogeneous polymer surface acts as an extra phase where the medium needs to permeate through the polymeric barrier to trigger the diffusion, and once the process starts, the diffusion may be delayed in comparison to pure MBGN [21,37]. The combination of these effects results in a difference in the release kinetics between the composite microspheres and the MBGN carriers.

Nevertheless, both systems exhibited a proper potential for the ASX-controlled release applications, and this behaviour may be extended to other phytotherapeutics and other applications in which high initial concentrations must be avoided [21,49,50]. Additionally, a delayed release may be useful in applications that require a time gap before the full release of the entrapped compound, such as imagining diagnostics applications, among others [32]. Possibly, the variation in the MBGN concentration in the composite microspheres may bring a pathway to tailor the release kinetic and achievement of prolonged release.

### 3.4. Bioactivity Assessment

After 14 days of immersion in Simulated Body Fluid (SBF), the structure and morphology of MBGN/ASX and composite microspheres were further analysed via XRD and SEM observation (Figure 4a and Figure 4b, respectively). In both graphs, it is possible to observe XRD patterns, namely, two peaks at 2θ values of 29° and 32°, characteristic of the formation of hydroxyapatite (HA) [21,51]. It is expected that in the MBGN/ASX and composite microspheres, calcium ions released from the systems interact with the ions present in SBF, promoting hydroxyapatite mineralisation on the surface. This HA layer was also confirmed via SEM observation (Figure 4c,d), where the characteristic morphology of HA was present all over the surface of both particles. Similar morphology has been observed in other BG-based structures after immersion in SBF [21,52]. These results support the bioactivity of both ASX/MBGN and the composite microsphere systems.

### 3.5. Cell Viability of Particle Systems

The cell viability of both MBGN/ASX and the composite microspheres administrated to NIH-3T3 cells was assessed after 24 and 72 h; blank MBGN, blank PHBV particles, and free ASX were used as additional conditions, and cells without treatment were considered as control (shown in Figure 5). The ASX amount was calculated to have a similar final concentration in both particle systems (4 μg/mL approx.), the same as the release kinetics assay. In the WST-8 cell assay after 24 h (Figure 5a), it was possible to observe that all conditions exhibited cell viability similar to the control, and only some dilutions of free ASX (10^−1^), MBGN/ASX (10^−2^ and 10^−3^), and composite microspheres (10^−1^) presented significant differences with respect to the control. On the other hand, after 72 h of incubation (Figure 5b), some dilutions of MBGN blank (10^−1^ and 10^−2^) and MBGN/ASX (10^−1^) exhibited a significant decrease in cell viability, while the composite microspheres (10^−1^ and 10^−2^) sustained an increase in cell viability. At this time, free ASX and blank PHBV presented no significant difference with respect to the control.

Fluorescent microscopy observations after 72 h (Figure 5c) corresponded to WST-8 cell viability results, in which it was possible to observe a decrease in the population of living cells on the samples treated with MBGNs and PHBV blank particles. This behaviour has been reported before due to the accumulation of high concentrations of nanoparticles on cells that may increase the difficulty for the assimilation of the particles and modify the cell viability as a consequence [21,53]. Nevertheless, cell viability values for all conditions were above 70%, which may indicate good cytocompatibility of this category of devices [38,46]. On the other hand, it was noticed that the administration of both free ASX and the ASX released from the composite microspheres promoted sustained cell viability after 72 h, where the antioxidant properties of ASX might support this behaviour by reducing the oxidative stress due to the contact and assimilation of the particles [5,6]. However, this behaviour was not observed in the MBGN/ASX system, which may be explained based on the release kinetic of this system (Figure 3b), whereby a higher release ratio compared to the composite microspheres may result in a faster depletion of the loaded ASX, which during the first hours helps in damping the side effects due to the assimilation of MBGN. Nevertheless, as previously mentioned, the promoted cell viability cannot be sustained after the full release of ASX. On the other hand, the prolonged release through the composite microspheres may extend the beneficial effects of the phytotherapeutic up to 72 h, as a similar population of living cells was observed for the samples treated with free ASX and the composite microspheres.

The results mentioned above show the importance of the release kinetics for the effective administration of the model drug by the two studied delivery systems. Another important aspect was the difference in the EE of both systems, indicating that a higher amount of MBGN/ASX is required to release equivalent amounts of ASX for extending the drug’s beneficial effects. Nonetheless, this also will increase the amount of MBGN in contact with cells, which, as discussed before, may impact biocompatibility. Furthermore, it could be interesting to characterise the influence of the MBGN content in the composite release kinetic as an easy alternative for tailoring the release in these kinds of carriers to determine if this behaviour can be extended to other model drugs. In any case, both the MBGN/ASX and composite microspheres exhibited an optimum potential for controlled phytotherapeutic delivery needed for specific applications.

## 4. Conclusions and Outlook

FTIR measurements confirmed the successful incorporation of ASX in MBGNs and composite microspheres. The MBGN/ASX and composite microspheres exhibited an optimum ASX entrapment efficiency (over 60% and 80%, respectively). The release kinetic curves showed a significant difference between the release behaviour of the MBGN/ASX and composite microspheres, where the latter exhibited prolonged release of ASX up to the first 80 h compared to half (40 h) of the MBGN/ASX.

The WST-8 cell viability results and fluorescent cell morphology observations showed that the difference in the release behaviour affected the NIH-3T3 cell viability during the first 72 h of incubation. These results suggested that the release behaviour of the composite microspheres may be tuned by modifying the MBGN content in the composite microspheres. Additionally, the exhibited release kinetics may be extended to other phytotherapeutics similar to ASX.

Although the composite microspheres are promising for sustained in vitro drug release applications, additional characterisation on these systems could be interesting, such as the effect of varying the MBGN content on the drug release kinetics and the stability and antioxidant activity of ASX during the particle synthesis process and the release.

## Figures and Tables

**Figure 1 polymers-15-02432-f001:**
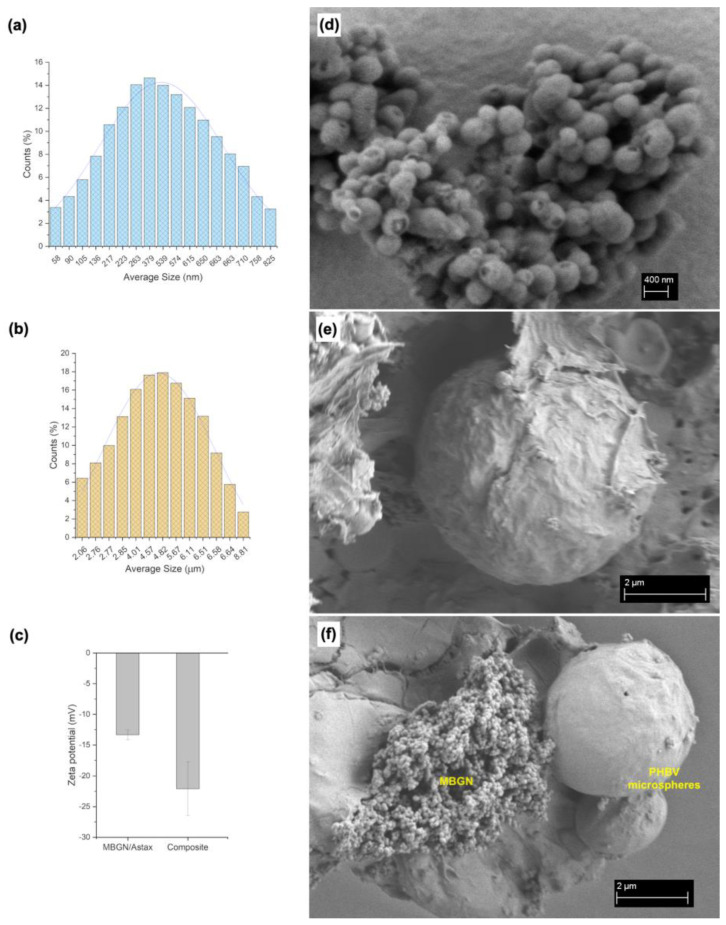
Particle size distribution of (**a**) MBGN/ASX and (**b**) composite microspheres; (**c**) Zeta potential of MBNG/ASX and composite particles. SEM micrographs showing the surface morphology of (**d**) MBGN/ASX and (**e**) composite microspheres and (**f**) comparison of MBGN and composite particle dimensions.

**Figure 2 polymers-15-02432-f002:**
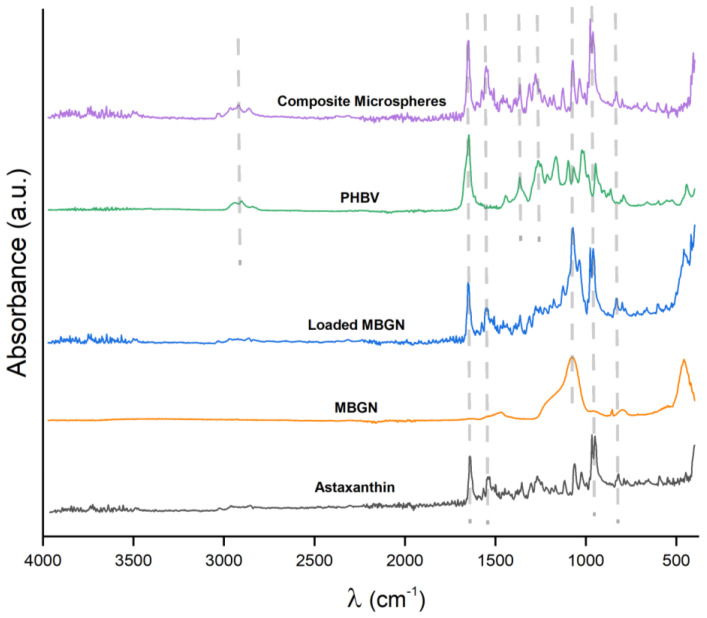
FTIR spectra of ASX, MBGN, MBGN/ASX, blank PHB, and composite microspheres. (λ is wavenumber).

**Figure 3 polymers-15-02432-f003:**
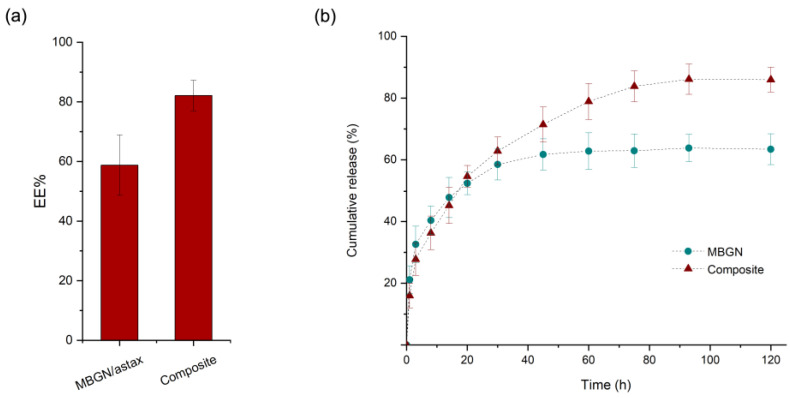
(**a**) ASX Entrapment Efficiency and (**b**) cumulative release kinetic of MBGN/ASX and composite microspheres (SD as error bars, *n* = 3).

**Figure 4 polymers-15-02432-f004:**
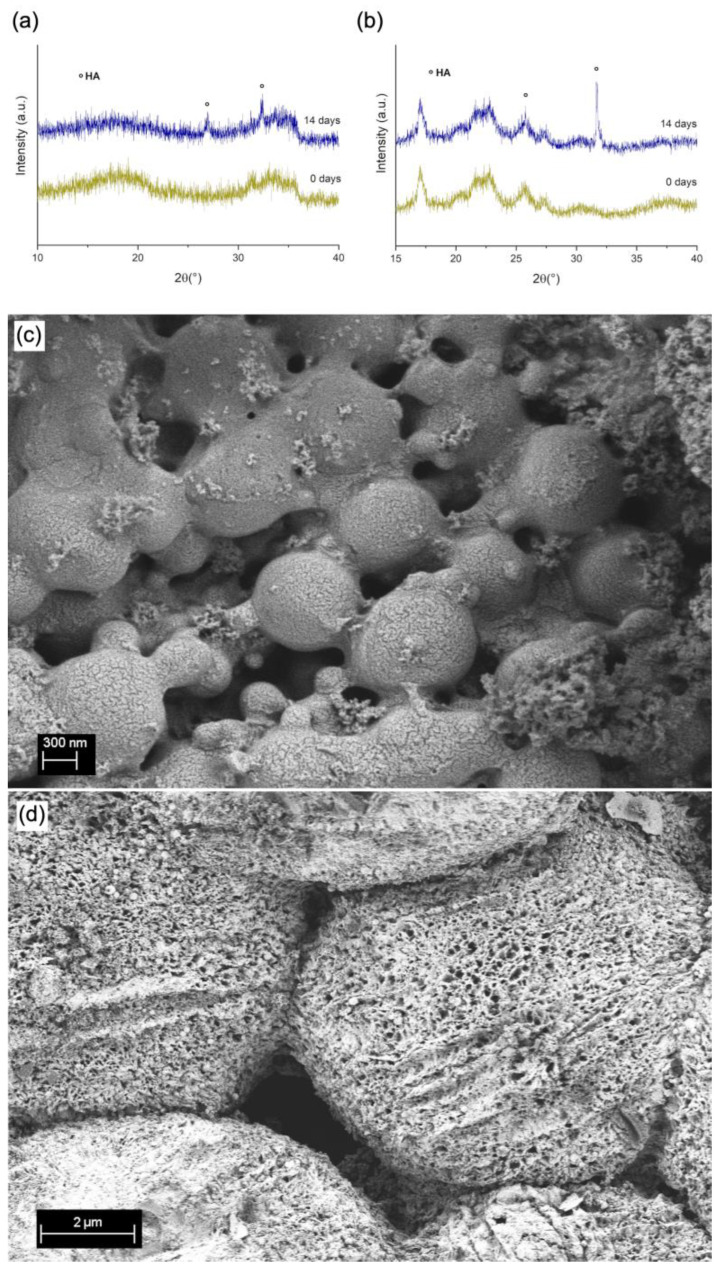
XRD patterns of (**a**) MBGN/ASX and (**b**) composite microspheres. SEM micrographs of (**c**) ASX/MBGN and (**d**) composite microspheres after 14 days of immersion in SBF.

**Figure 5 polymers-15-02432-f005:**
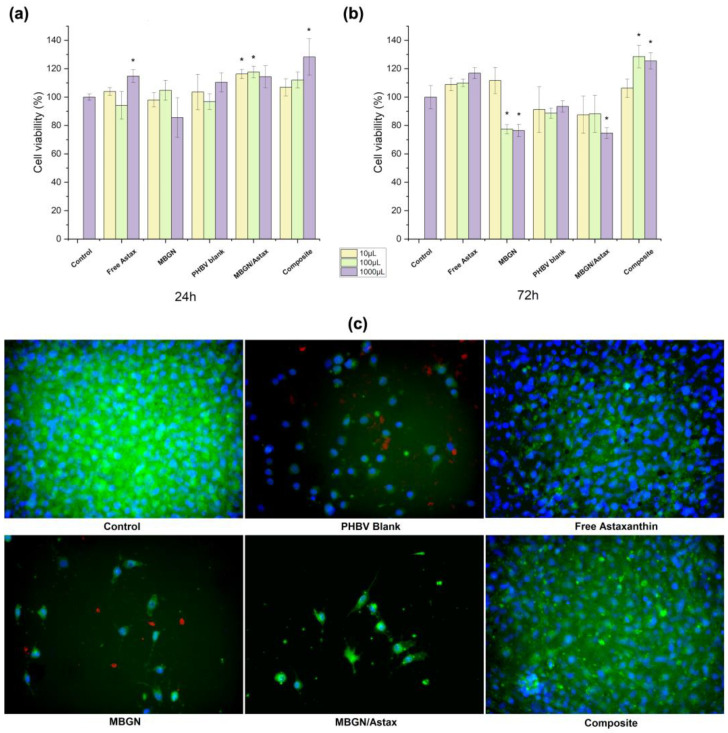
Cell viability of NIH-3T3 cells treated with the composite microspheres, MBGN/ASX, blank PHBV microspheres, blank MBGN, free ASX, and cells without any treatment as control after (**a**) 24 h and (**b**) 72 h of incubation; (**c**) fluorescence microscopy micrographs of each condition after 72 h. (* indicates a significant difference of *p* ≤ 0.5 Tukey test, SD as error bars, *n* = 3).

## Data Availability

Data is available from the author.

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
