# Peer review of "Comparison between the Astaxanthin Release Profile of Mesoporous Bioactive Glass Nanoparticles (MBGNs) and Poly(3-hydroxybutyrate-*co*-3-hydroxyvalerate) (PHBV)/MBGN Composite Microspheres"

_polymers, 2023, doi:10.3390/polym15112432_

Round 1

Reviewer 1 Report

Dear Author,

    The work was designed well and surely this work would be a great reference for the future researchers in this focus area. However very few minor correction are found in the manuscript, the author may overcome by a proof reading.

A major correction suggested in the chapter division. kindly modify a follows

1. Introduction

2. Materials and methods

3. Results and discussion

4. Conclusion

The other correction were attached in the file attached, please go through and do the neeful. 

Thank you

A minor revision requested

Author Response

Dear Reviewer,

Derived from the comments you kindly communicated, the following modifications have been endorsed to the manuscript:

  1. The sections of the manuscript have been re-ordered according to the suggestion: Introduction, Material and Methods, Results and Discussion, Conclusions, etc.
  2. All the comments and orthographical suggestions have been corrected (see tracking changes).

We hope these modifications are sufficient to have your acceptance of this version of the manuscript for publication.

Best regards,

The Authors

Reviewer 2 Report

 In the manuscript, Astaxanthin (ASX) was loaded into the mesoporous bioactive glass nanoparticles (MBGN), poly (3-hydroxybutyrate-co-3-hydroxyvalerate) (PHBV) microspheres and MBGN/PHBV composite material. The authors compared ASX release kinetic, ASX entrapment efficiency and cell viability in both MBGN nanoparticles, PHBV microspheres and MBGN/PHBV composite material.  It was claimed that the composite microspheres exhibited a more prolonged release profile with sustained cytocompatibility.

This is an interesting study. However, the provided data sets to support the conclusion is not sufficient and thus it seems premature to proceed with the manuscript based on the current results. Hence, I recommend revision of manuscript to refine.

1.       The authors should add the ratio of MBGN/PHBV composite material used for all experiments.

2.       Transmission electron microscopy images should be added for displaying encapsulation of ASX by particles.

3.       The authors should add XRD data for day 7 and 21.

4.       The author should add more cell viability data sets with different concentrations such as 200, 500, 700 μL addition to 10, 100 and 1000 μL in Figure 5 a, b.

5.       Fluorescence microscopy micrographs are not clear. Authors should add better quality images.

Minor editing of English language required.

Author Response

Dear Reviewer,

Derived from the comments you kindly communicated, the following modifications have been endorsed to the manuscript, including the answers to the questions that were made, and they are described below:

  1. The MBGN/PHBV composite material ratio has been clarified in the Materials and Methods section (see tracking of changes).
  2. The experimental part of this work was completed by the beginning of 2022 as part of a one-year postdoc position. Currently, a few of the authors are conducting research in different institutions. Unfortunately, due to these reasons is not possible to do any further characterization such as Transmission Electron Microscopy.
    However, the presence of Asthaxantin on the synthesized particles used for the experiments is supported by the FTIR spectra and the UV/vis entrapment efficiency results.
  3. In previous works (cited in the manuscript) has been reported that in these kinds of systems (PHBV/MBGN), the formation of a layer of hydroxyapatite on the surface is detectable by XRD at 7 days of SBF immersion and completely observable after the 14 days [21, 24, 35]. Moreover, SEM micrographs showed the typical hydroxyapatite structure on the microspheres after 14 days. The reason why the assay was stopped at 14 days. 
  4. Due to the same reasons as the previous point, no further experiments are possible. Nevertheless, teen times dilution is a good range to show the cell viability behaviour, and this exponential dilution has been reported previously [21, 32, Idris M et. al. 2018].
  5. The high-resolution image is uploaded and will be available in the online version of the manuscript, we attached a high-res JPG image below as an example.

Additionally, orthographic errors were corrected and double-checked English,  the sections of the manuscript have been re-arranged to better lecture also. 

We hope these modifications are sufficient to have your acceptance of this version of the manuscript for publication.

Best regards,

The authors

Round 2

Reviewer 2 Report

I recommend this manuscript for the publication.